# Consumption of Aged White Wine under a Veil of Flor Reduces Blood Pressure-Increasing Plasma Nitric Oxide in Men at High Cardiovascular Risk

**DOI:** 10.3390/nu11061266

**Published:** 2019-06-04

**Authors:** Irene Roth, Rosa Casas, Margarita Ribó-Coll, Ramón Estruch

**Affiliations:** 1Department of Internal Medicine, Institut d’Investigacions Biomèdiques August Pi Sunyer (IDIBAPS), Hospital Clinic, University of Barcelona, 08036 Barcelona, Spain; ROTH@clinic.cat (I.R.); mribocoll@gmail.com (M.R.-C.); rcasas1@clinic.cat (R.E.); 2CIBER Fisiopatología de la Obesidad y Nutrición (CIBEROBN), Instituto de Salud Carlos III, (ISCIII), 28029 Madrid, Spain

**Keywords:** wine, Andalusian aged white wine, blood pressure, nitric oxide, cardiovascular risk, polyphenols

## Abstract

Background: Hypertension remains the largest attributable risk factor of cardiovascular disease (CVD), and a reduction of cardiovascular events is linked to diminished elevated blood pressure (BP) values. High alcohol intake is a common cause of hypertension, but some studies have suggested that moderate wine consumption may reduce BP and increase plasma nitric oxide (NO) due to its polyphenol content. Objective: The aim of the present study was to compare the effects of Andalusian aged white wine (AWW) under a veil of flor, an alcoholic beverage with a moderate polyphenol content, with those of gin, an alcoholic beverage without polyphenols, on BP and plasma NO in men at high cardiovascular risk. Methods: This study was designed as an open, randomized crossover-controlled trial in which 38 high-risk male volunteers, aged 55 to 80, received 30 g of ethanol daily in the form of AWW or gin. This was carried out over the course of three weeks, after a two-week washout period. At baseline and after each intervention period, BP, anthropometric parameters, and plasma NO were measured; food intake was also recorded, and physical activity was monitored. Results: Compared to gin, AWW significantly reduced systolic and diastolic BP (*p* ≤ 0.033; both) and increased plasma NO levels (*p* = 0.013). Additionally, changes in BP values observed after AWW significantly correlated with increases in plasma NO. No changes in food intake, physical activity, body weight, or waist were observed between the two intervention periods. Conclusions: Moderate daily consumption of AWW may be useful to reduce elevated BP due to an increase of NO synthesis. This effect could be attributed to grape-derived compounds in AWW, such as polyphenols, which are not present in gin.

## 1. Introduction

Elevated blood pressure (BP) represents a public health problem worldwide and is considered one of the major underlying causes of morbidity and mortality in cardiovascular disease (CVD) [1,2]. Epidemiological evidence suggests that moderate alcohol consumption may reduce CVD [3] and that alcoholic beverages may exert at least part of these beneficial effects by lowering BP [4].

Although hypertension is a multifactorial disease, there is growing scientific evidence showing that dietary nitrite and nitrate intake can affect BP values and even CVD [5]. Population studies have observed a negative association between dietary intake of nitrites and nitrates and the risk of hypertension [4,5,6,7,8,9]. This association has also been reported in short-term clinical trials evaluating the effects of nitrate-rich foods, such as apples and spinach, in hypertensive patients [7,10,11]. Similarly, interventional studies performed in experimental animals and humans have evaluated the effects of red wine extract, alcohol, and red wine on BP and reported an association between the lowering effect of red wine on BP and the increasing plasma concentrations of nitric oxide (NO) observed. The reduction of BP is due to increased vasorelaxation induced by enhancing NO synthase activity and NO production [12]. In another clinical trial, the hypotensive effects of de-alcoholized red wine were greater than those observed with regular red wine, attributing this effect to the polyphenols and other bioactive compounds contained in red wine [13]. Among other proposed mechanisms of plasma NO increases include improvements in endothelial function and vascular relaxation, while other mechanisms proposed for the increase in plasma NO include improvements in endothelial function and vascular relaxation [5].

Since no previous study has evaluated the effects of white wines on BP and plasma NO concentrations, this clinical trial was aimed at analyzing the short-term effects (21 days) of a sherry variety of Andalusian white wine (AWW), also called “AWW aged under a veil of flor”, on BP parameters and plasma NO concentrations in men at high cardiovascular risk.

## 2. Materials and Methods

### 2.1. Participants

Participants were men aged between 55 to 80 years with no previous history of documented CVD, who were screened from October 2012 to February 2014 in a primary care center associated with the Hospital Clínic of Barcelona (Spain). All participants were low-to-moderate alcohol consumers (around 30 g/ethanol/day) and had diabetes mellitus or 3 or more of the following CVD risk factors: active smoking, hypertension, plasma LDL cholesterol of >4.13 mmol/L (>160 mg/dL), plasma HDL cholesterol of <0.905 mmol/L (<35 mg/dL), overweight, or obesity (body mass index [BMI] of >25 kg/m²), and/or a family history of premature coronary heart disease (CHD). The exclusion criteria included documented CHD, stroke or peripheral vascular disease, human immunodeficiency virus infection, alcoholic liver disease, malnutrition, and neoplastic or acute infectious diseases. None of the study subjects were consumers of vitamin supplements or anti-inflammatory drugs (steroids, non-steroidal anti-inflammatory agents, or aspirin).

The Institutional Review Board of the Hospital Clinic approved the study protocol, and all participants gave written consent to participate in the study. Furthermore, this study was registered in the International Standard Randomized Controlled Trial Number (ISRCTN) register with the number 01319643 (www.controlled-trials.com).

### 2.2. Study Design

The study was an open, randomized, controlled, crossover trial with four intervention periods. Each intervention period was comprised of three weeks, with a 2-week washout period between the two interventions. During the washout period, the volunteers were asked not to consume any alcoholic beverages. Moreover, the design, methodology, and eligibility criteria for this study have been published previously [14].

After signing the informed consent form, volunteers were randomized using a computer-generated random-number table into one of two interventions in a crossover design; one included the administration of 92 mL of gin (30 g ethanol/day) and the other included the same amount of ethanol as AWW (255 mL total phenol: 927.79 Eq. gallic acid/day [EGA/day]), with an ethanol content of 13%. Both groups followed a common background diet. Furthermore, a washout period was included between interventions.

The phenolic profile of the AWW and gin used in the trial was determined by SPE-LC-ESI-MS/MS and is shown in Table 1. Interestingly, the total polyphenol compounds of AWW used in this study were nearly 3-fold the amount of polyphenols detected in a common white wine (321 milli Equivalent gallic acid (mEqGA)/L) [14]. No detectable phenolic compounds were observed in gin.

### 2.3. Diet and Exercise Monitoring

The participants were asked to maintain their usual dietary habits and physical activity. They were also asked to abstain from alcoholic beverages other than those provided by the investigative team. Participants were not blinded to the type of drink ingested. Furthermore, the participants’ diet was closely monitored, especially in relation to foods rich in antioxidants, such as fruit, vegetables, tea, and coffee, to ensure that the individual diets had similar antioxidant content throughout the study.

At baseline and after each intervention period, diet and physical activity were monitored. A validated 3-day (2 work days and 1 weekend day) food recall questionnaire was used to evaluate the diet followed by each participant during each intervention period [15]. These questionnaires were also used to assess adherence to the study. The dietary information recorded was converted into nutrient data using the Food Processor Nutrition and Fitness Software (ESHA Research, Salem OR, 2012 10.10.0), which was adapted to local foods. Furthermore, the Minnesota Leisure Time Physical Activity Questionnaire was administered to monitor the physical activity of the participants throughout the study [16]. Empty AWW and gin bottles of the participants were counted to ensure compliance. 

At the end of each intervention period, a checklist of clinical symptoms such as bloating, altered bowel habit, or dizziness, as well as other possible symptoms, was administered to the participants to identify possible side effects related to the interventions.

### 2.4. Clinical and Laboratory Measurements

At baseline and after each intervention, a trained nurse measured the BP and heart rate in triplicate at 5 min intervals in the non-dominant arm after 15 min of resting in a seated position. The BP was measured with a validated semiautomatic oscillometer (Omron 705 CP; Omrom Matsuasaka Co Ltd., Matsuasaka City, Japan). The mean of the second and the third measurements was used for statistical analysis. In addition, anthropometric measurements including height, body weight, waist, and hip perimeters were made using standardized methods [17].

The last day of the run-in period (at baseline) and after each intervention period (AWW or gin) fasting blood, serum, EDTA-plasma, and urine samples were collected and immediately centrifuged and stored at −80 °C until assayed. The participants had fasted 12 h prior to blood analyses. Additional serum analytes including plasma aspartate aminotransferases (AST), alanine transaminase (ALT), gamma glutamyltranspeptidase (GGT), albumin, cholesterol, prothrombin time, folic acid, and vitamin B_12_ were measured to determine the possible adverse effects of ethanol intake. For the measurement of nitric oxide (NO), the release of NO_2_^–^ and NO_3_, the stable breakdown products of NO in plasma, were determined by a chemiluminescence detector in an NO analyzer (Sievers Instruments, Boulder, CO, USA) [13]. Finally, the concentration of tartaric acid in 24 h urine was also measured before and after each intervention to assess compliance with the intervention assigned [18].

### 2.5. Statistical Analysis

The ENE 3.0 statistical program (GlaxoSmithKline, Brentford, United Kingdom) was used to determine sample size. For a cross-over design and assuming a maximum loss of 10% of participants, to detect a mean difference of 4.1 µmol/L of NO with a conservative standard deviation (SD) of 6.0, at least 22 subjects are needed for the study (α risk = 0.05, power = 0.9).

The statistical analyses were carried out using the SPSS statistical analysis system v20.0 (SPSS, Chicago, IL, USA). Descriptive statistics (mean ± SD or n [%]) were used to describe the baseline characteristics of the participants and the outcome variables. The paired two-tailed t-test was used to compare changes in outcome variables in response to each intervention period and carryover effects for the outcome variables observed before the AWW and gin periods. We also compared the differences in the parameters obtained between the groups starting with AWW and then with gin. The paired two-tailed t-test was also used to compare differences in the effects of each intervention. Pearson’s correlation analysis was used to quantify relationships between changes in blood pressure and plasma NO concentrations. The effects of each intervention, as well as differences between the two interventions, are expressed as mean changes (95% confidence interval, CI).

## 3. Results

### 3.1. Characteristics of the Participants and Assessment of Side Effects

A total of 47 potential candidates were pre-screened for eligibility. Figure 1 shows the study flowchart. Of the 47 candidates, two did not meet the inclusion criteria and four refused to participate; therefore, 41 were randomized to two sequences of interventions: (1) AWW and gin (*n* = 21); and (2) Gin and AWW (*n* = 19). However, three participants included in the first sequence dropped-out; thus, 38 completed the study and were included in the analysis. Table 2 shows the baseline characteristics of the participants studied. Most were overweight or obese (88%), more than half had dyslipemia (54%), nearly three-quarters had hypertension (73%), and a fifth have type-2 diabetes (21%), while 13% were smokers.

Serum folic acid, vitamin B12, albumin, creatinine, AST, ALT, and GGT concentrations remained within the normal range throughout the study. No subjects reported any adverse effects during the interventions.

### 3.2. Intervention, Diet, and Physical Activity Monitoring

In a crossover design, all participants underwent both interventions—30 g of ethanol as AWW or gin in a random sequence. Twenty-one participants started with AWW and the other 19 with gin. The number of empty bottles returned after each intervention period, and the results of the self-reported 3-day questionnaire of food consumption, showed good adherence to the study protocol. Tartaric acid was measured as a biomarker of wine consumption and was also used as a measurement of intervention compliance [18]. This measurement was made in 24 h urine samples that were collected on the last day of the run-in period and after each intervention. Results showed a significant increase in the 24 h excretion of tartaric acid after AWW consumption (from 14.09 ± 31.80 to 63.79 ± 39.73 µg/mL, *p *< 0.001).

No changes in the usual dietary habits or physical activity were reported throughout the study in the participants’ dietary and Minnesota reports (Table 3). In addition, none of the participants reported changes in the medications used.

### 3.3. Changes in Cardiovascular Risk Factors after 3 Weeks of Intervention

After the 3-week intervention with AWW, systolic and diastolic BP showed a mean reduction of −4.91 mmHg (95% CI: −9.41 to −0.42; *p *= 0.033) and −2.90 mmHg (CI: −5.50 to −0.29; *p *= 0.030), respectively. A comparison between groups showed significant reductions in systolic BP and diastolic BP after AWW intake compared to gin (*p *< 0.040; both) (Table 4). Following the AWW intervention, plasma NO concentrations significantly increased from baseline to 21 days after the intervention (*p *= 0.013); however, no significant changes were observed in a comparison between groups (*p *> 0.05).

In the subgroup analyses, we did not find differences in the effects of AWW and gin on BP and plasma NO concentrations in the different subgroups analyzed; that is, in smoker vs. non-smoker, diabetic vs. non-diabetic, hypertensive vs. non-hypertensive, and dyslipidemic vs. non-dyslipidemic subjects.

## 4. Discussion

After 21 days of intervention with AWW and gin, in a crossover study on high cardiovascular risk subjects, only AWW decreased systolic and diastolic BP and increased the plasma NO concentrations. This suggests that the hypotensive effects of AWW should be attributed to non-alcohol compounds of this type of wine, which may induce vasodilatation due to increases in plasma NO concentrations. In this cross-over study, men with high cardiovascular risk followed two 3-week interventions with AWW and gin with two 15-day washout periods before each intervention. When comparing the results of AWW and gin intake, systolic and diastolic BP decreased and plasma NO concentrations increased only after the AWW intervention, suggesting that the hypotensive effects of AWW might be attributed to the non-alcohol compounds of this type of wine, which may induce vasodilatation due to an increase in plasma NO concentrations.

For many years, NO has been considered an attractive therapeutic target for the cardiovascular system. Indeed, this interest stems from various lines of research showing that increased NO concentrations lead to vasodilatation and reduced BP, as well as improving arterial stiffness after dietary nitrate consumption [7,10,19]. Currently, a large body of scientific evidence supports the role of NO as the key regulator of vascular homeostasis and as a natural vasodilator, because NO reduces the vascular oxidative stress and inflammation associated with arterial aging, making it a potential therapeutic option in CVD [20,21,22,23,24].

In fact, systemic inflammation and oxidative stress are the pathophysiological bases of atherosclerosis and cardiovascular risk factors [25]. We have previously reported [14] that moderate consumption of AWW significantly improves systolic and diastolic BP, as well as HDL-cholesterol and apolipoprotein A1 concentrations. While changes observed in HDL and apolipoprotein A1 concentrations might be attributed to alcoholic fraction (ethanol), the BP-lowering effects also observed in the study could be attributed to polyphenols (such as resveratrol or others) or other minor components of AWW and not to alcohol. Additionally, we also reported a reduction in the expression of leucocyte adhesion molecules, circulating endothelial progenitor cells, and inflammatory cytokines and chemokines related to atherosclerosis after consumption of AWW. Since these effects were not observed after gin intervention, we attributed them mainly to polyphenols in AWW. In this manuscript, we also underlined that no side effects were observed in both intervention arms, including in liver and kidney function tests [14].

Other studies have examined the relationship between hypertension and oxidative stress in experimental animals [26] and hypertensive subjects [27,28]. However, the results of these studies concluded that treatment with antioxidants is ineffective in hypertension, suggesting that oxidative stress is an effect (a consequence) rather than the cause of essential hypertension [26].

Heavy alcohol intake increases the risk of hypertension in a dose-dependent manner. However, to date, the beneficial effects observed between light or moderate alcohol consumption and hypertension is controversial because of the confusing trends and results observed. Indeed, while some studies have reported a linear relationship between alcohol consumption and its benefits, others did not find this linearity or even show a J-shaped association [29,30]. In a recent meta-analysis [31] including a total of 16 prospective studies (33,904 men and 193,752 women), it was found that the risk of hypertension in men not only increased with heavy alcohol consumption (31 to 40 g/day) but also with low (<10 g/day) and moderate (11 to 20 g/day) alcohol consumption. On the other hand, women showed a J-shaped relationship between alcohol consumption and hypertension. By contrast, another meta-analysis and systemic review based on 20 cohort studies (125,907 men, 235,347 women, and 90,160 incident cases of hypertension) found no evidence for a protective effect of alcohol consumption on hypertension in women [32]. In addition, Taylor et al. [33] concluded that the risk of hypertension is dose-dependent with alcohol consumption, but this dose-response relationship starts after an apparent threshold effect at two standard drinks (24 g pure alcohol per day). With lower doses, there was an apparent lowering effect on BP [34]. According to the results obtained in the current study and another by Chiva-Blanch et al. [13], the final effect of an alcoholic beverage on BP depends on the alcohol strength of the beverage and the amount of polyphenols that it contains.

Additionally, numerous studies on the effects of polyphenols [35,36] have shown reductions in inducible nitric oxide synthase (iNOS) expression, which is considered to be an anti-inflammatory effect of polyphenols [36] and activation of endothelial nitric oxide synthase (eNOS), which has positive effects on BP [37]. Several in vitro animal and experimental studies have demonstrated that moderate consumption of wine, mainly red wine, leads to lower BP values and the enhancement of endothelial nitric oxide production [38]. Nevertheless, although it has been reported that wine phenolic compounds may exert their cardioprotective effect by lowering endothelial dysfunction and increasing NO production and its availability [39,40], scientific evidence of the effects of wine polyphenols on BP in humans remains inconsistent [41,42,43]. In a double-blind, placebo-controlled three-period crossover trial [44], intake of red wine polyphenols for four weeks did not reduce BP in subjects with high-normal BP or grade one hypertension. Similar results were found in another study performed in healthy young women [45]. Contrarily, Sivaprakasapillai et al. [46] randomized participants with metabolic syndrome into three parallel groups to consume a placebo, 150 mg grape seed extract/d, or 300 mg grape seed extract/d during a 4-week treatment period. Both systolic and diastolic BP decreased after treatment with the grape seed extracts compared with the placebo. Moreover, our results showed significant reductions in BP after moderate consumption of AWW in subjects at high risk of CVD, who most likely had endothelial dysfunction. The mechanism associated with a reduction of BP might be due to the intake of the phenolic compounds (resveratrol and probably others) present in AWW. These polyphenols would increase eNOS expression, improving endothelial dysfunction, and reducing BP. Some studies have also shown that high doses of resveratrol (≥150 mg/d) reduce systolic BP [47].

On the other hand, Huang et al. [3] reported a significant increase in plasma NO concentrations in healthy participants who had consumed 100 mL of red wine daily for three weeks, though no changes were observed in BP values when volunteers consumed vodka or beer with an equivalent amount of alcohol. Barona et al. [48] also demonstrated that compared with a placebo, the daily consumption of grape extract for 30 days significantly improved vascular endothelial function and biomarkers of metabolic syndrome by increasing flow-mediated vasodilation response and lowering systolic BP. As expected, the participants maintained their usual body weight, diet, and physical activity throughout the whole study. In addition, the consumption of a grape preparation with standardized polyphenol content led to an improvement in vascular endothelial dysfunction, increasing NO bioavailability in volunteers, aged 30–70 years, with metabolic syndrome [48]. As in the case of BP, the plasma levels of NO in our study also improved following the AWW intervention, probably by polyphenols inducing iNOS production as previously shown in some experimental studies [49]. Chiva-blanch et al. [13] confirmed this hypothesis in their study, showing that de-alcoholized red wine polyphenols decreased BP while increasing NO production of iNOS in healthy humans.

This study has some limitations. On the one hand, we did not identify the polyphenol(s) responsible for the effects observed. In addition, endothelial function was not measured. Furthermore, it would be interesting to also analyze oxidative stress markers such as superoxide dismutase (SOD), glutathione peroxidase (GPx), or NADPH oxidase to confirm the results observed in relation BP or NO levels. Finally, on the other hand, a 3-week intervention may not represent the potential effects of long-term consumption.

## 5. Conclusions

This feeding trial suggests that moderate consumption of AWW has beneficial effects on the cardiovascular system by increasing plasma NO, which may contribute to decreasing the risk of developing hypertension and, consequently, diminishing cardiovascular risk.

## Figures and Tables

**Figure 1 nutrients-11-01266-f001:**
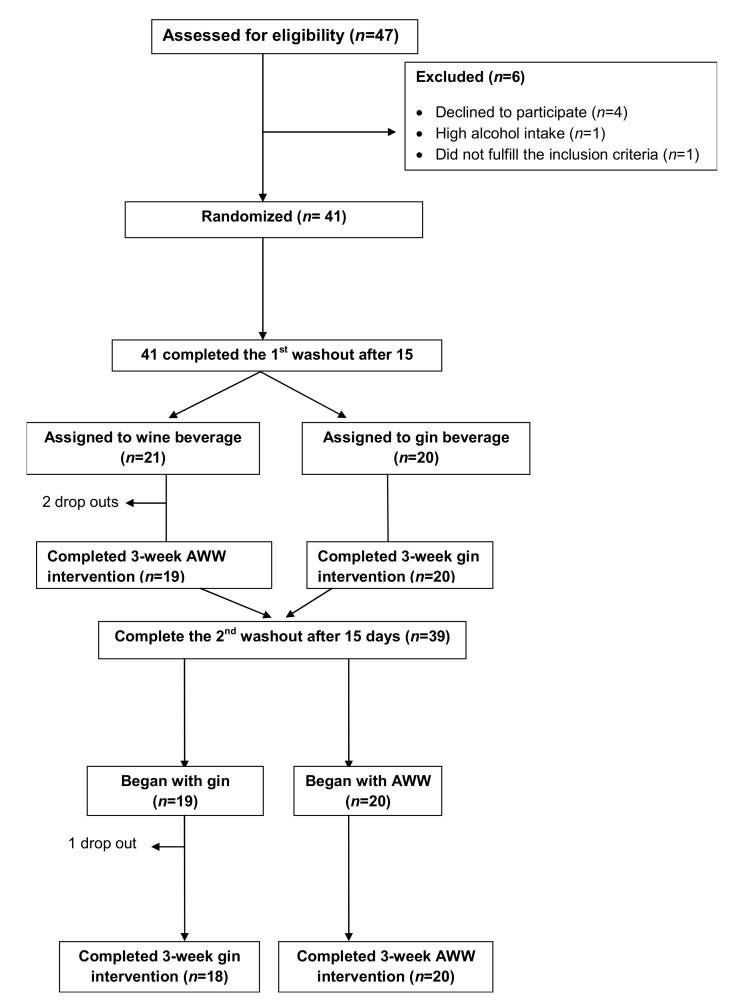
Flowchart.

**Table 1 nutrients-11-01266-t001:** Phenolic composition of the aged white wine studied.

	AWW
**Total Phenols (mEqGA/L)**	927.8
Gallic acid (mg/L)	6.0
Protocatechuic acid (mg/L)	5.6
4-Hydroxybenzoic acid (mg/L)	0.3
Vanillic acid (mg/L)	0.7
Syringic acid (mg/L)	1.5
Caffeic (mg/L)	1.0
Caffeoyl tartaric acid(mg/L)	17.5
Ferulic acid (mg/L)	0.2
p-Coumaric acid (mg/L)	0.2
p-Coumaroyl tartaric acid(mg/L)	24.2
**Flavanols**	
Catechin (mg/L)	23.7
Epicatechin (mg/L)	12.5
Procyanidin dimer B1 (mg/L)	20.0
Procyanidin dimer B2 (mg/L)	7.2
Procyanidin dimer B3 (mg/L)	14.7
Procyanidin dimer B4 (mg/L)	3.5
**Flavonols**	
Quercetin (mg/L)	0.1
Quercetin 3-*O*-arabinoside (mg/L)	-
Quercetin 3-*O*-rutinoside (mg/L)	-
**Other Polyphenols**	
Tyrosol (mg/L)	57.3
Vanillin (mg/L)	2.2
Resveratrol (mg/L)	-
*trans*-Resveratrol (mg/L)	-
*cis*-Resveratrol (mg/L)	-

AWW: Aged White Wine; GAE: Gallic Acid Equivalents.

**Table 2 nutrients-11-01266-t002:** Baseline characteristics of the 38 study subjects.

	Mean ± SD or %
Age (years)	68.7 ± 5.4 ^a^
Height (cm)	168.9 ± 6.2
Body weight (kg)	82 ± 11.1
BMI (kg/m²)	28.7 ± 3.2
BMI ≥ 25 kg/m² [n (%)]	35 (92) ^a^
Waist circumference (cm)	105 ± 7
Hypertension [n (%)]	27 (71)
Type 2 diabetes mellitus [n (%)]	8 (23)
Dyslipidemia [n (%)]	20 (53)
Current smokers [n (%)]	5 (13)
Sedentariness [n (%)]	4 (11)
Family history of premature CHD [n (%)]	4 (11)
Systolic blood pressure (mmHg)	144.2 ± 16.2
Diastolic blood pressure (mmHg)	76.9 ± 9.7
Heart rate (beats/min)	65.3 ± 9.2
Diuretics [n (%)]	10 (26)
Statins [n (%)]	15 (39)
Fibrates [n (%)]	13 (34)
ACE inhibitors [n (%)]	19 (50)
Oral hypoglycemic drugs [n (%)]	14 (37)
Aspirin or antiplatelet drugs [n (%)]	9 (24)
Others [n (%)]	17 (45)
Glucose (mg/dL)	111.3 ± 23.3
Total cholesterol (mg/dL)	170.8 ± 38.5
LDL-cholesterol (mg/dL)	103.9 ± 36.1
HDL-cholesterol (mg/dL)	44.8 ± 14.4
Triglycerides (mg/dL)	105.5 ± 54.5
ALT (IU/L)	25.5 ± 11.6
AST (IU/L)	25.3 ± 11.7
GGT (IU/L)	21.4 ± 13.8
Folic acid (serum) (ng/mL)	8.9 ± 2.9
Vitamin B12 (pg/mL)	399.5 ± 175.4
Apo A1 (mg/dL)	113.7 ± 18.9
Apo B (mg/dL)	79.3 ± 19.2
Lipoprotein (a) (mg/dL)	37.1 ± 54.1

SD: standard deviation; BMI: body mass index; CAD: coronary artery disease; ACE: angiotensin-converting enzyme; LDL: low-density lipoprotein; HDL: high-density lipoprotein; ALT: alanine aminotransferase; AST: aspartate aminotransferase; GGT: gamma glutamyl transpeptidase; ApoA1: apolipoprotein A1; ApoB: apolipoprotein B; ^a^ Values are mean ± SD or n (%).

**Table 3 nutrients-11-01266-t003:** Values of the mean nutrients and physical activity before and after the 3-week aged white wine (AWW) and gin interventions.

	Before AWW Mean ± SD ^+^	After AWW Mean ± SD ^+^	Mean Differences (95% CI) ^ǂ^	Before Gin Mean ± SD ^+^	After Gin Mean ± SD ^+^	Mean Differences (95% CI) ^ǂ^	*p*-Value
**Physical activity (METS/week)**	6560 ± 6333	6686 ± 6531	125.94 (−733.80 to 985.70)	5783 ± 4624	6477 ± 6379	694 (−1294to 2682)	0.481
**Energy (Kcal)**	1753 ± 264	1965 ± 353	212.03 (83.95 to 340.10) *	1730 ± 301	1972 ± 397	242.02 (−90.50 to 393.54) *	0.003
**Protein (g)**	82.35 ± 15,63	86.07 ± 14.61	3.71 (−2.75 to 10.18)	84.15 ± 16.53	83.61 ± 18.01	−0.53 (−8.94 to 7.86)	0.896
**Carbohydrates (g)**	187.72 ± 39.05	196.64 ± 50.72	8.92 (−6.63 to 24.48)	187.81 ± 47.92	196.53 ± 54.37	8.71 (−14.96 to 32.39)	0.456
**Dietary fiber**	17.19 ± 5.10	17.67 ± 6.96	0.48 (−1.81 to 2.78)	17.28 ± 6.25	16.90 ± 7.61	−0.38 (−3.24 to 2.47)	0.785
**Total fat (g)**	78.41 ± 17.82	73.01 ± 16.17	−5.41 (−12.54 to 1.72)	76.55 ± 18.28	73.30 ± 18.22	−3.25 (−11.71 to 5.20)	0.436
**Cholesterol (mg)**	278.42 ± 63.70	281.69 ± 106.31	3.26 (−44.32 to 50.86)	281.37 ± 72.40	272.48 ± 85.56	−8.89 (−54.70 to 36.92)	0.693
**Polyphenols (mg)**	1192 ± 374	1428.41 ± 641.62	236.30 (28.80 to 443.80) *	1281 ± 588	1294 ± 608	12.99 (−282.69 to 308.68)	0.929
**Alcohol (g)**	0.00 ± 0.00	26.17 ± 0.32	26.17 ( 26.04 to 26.30) *	0.00 ± 0.00	29.26 ± 1.88	29.26 (28.49 to 30.02) *	<0.001

Results are expressed as ^+^ mean ± standard deviation (SD) (*n* = 38) and ^ǂ^ mean differences (95% confidence interval [CI]) between before and after each intervention. Before each intervention is the value of the previous intervention or the baseline in the first intervention. METS: metabolic equivalent. * *p*: Significant differences (*p* < 0.05) between before and after the intervention (intra-group changes). *p*-Value: Statistical differences between group changes.

**Table 4 nutrients-11-01266-t004:** Mean biomarker values before and after the 3-week aged white wine and gin interventions.

BIOMARKERS	BEFORE WINE Mean ± SD ^+^	AFTER WINE Mean ± SD ^+^	Mean differences (95%Cl) ^ǂ^	*p*	BEFORE GIN Mean ± SD ^+^	AFTER GIN Mean ± SD ^+^	Mean differences (95% Cl) ^ǂ^	*p*	*p*-Value
Systolic BP (mmHg)	141.78 ± 14.94	136.87 ± 13.17	−4.91 (−9.41 to −0.418) *	0.033	137.43 ± 15.87	139.55 ± 17.51	2.12 (−2.31 to 6.56)	0.338	0.039
Diastolic BP (mmHg)	76.34 ± 9.29	73.44 ± 10.50	−2.90 (−5.50 to −0.295) *	0.030	72.83 ± 11.87	73.44 ± 910.87	0.61 (−1.64 to 2.86)	0.585	0.039
Heart rate (lpm)	65.15 ± 8.64	64.66 ± 9.69	−0.49 (−3.52 to 2.53)	0.743	65.76 ± 10.26	65.91 ± 9.45	0.15 (−3.26 to 3.57)	0.928	0.538
Weight (Kg)	81.77 ± 10.94	82.06 ± 10.64	0.28 (−0.21 to 0.78)	0.248	82.00 ± 10.86	82.29 ± 10.66	0.29 (−0.76 to 0.66)	0.116	0.496
Body mass index (Kg/m^2^)	28.73 ± 3.12	28.84 ± 3.04	0.11 (−0.07 to 0.30)	0.222	27.91 ± 5.15	28.73 ± 3.02	0.81 (−0.84 to 2.48)	0.325	0.285
Waist circumference (cm)	103.37 ± 7.61	102.98 ± 8.30	−0.38 (−1.70 to 0.92)	0.551	102.79 ± 8.06	102.95 ± 7.39	0.16 (−0.56 to 0.90)	0.648	0.484
NOx serum (µmol/L)	28.50 ± 12.62	56.37 ± 62.98	27.86 (−6.86 to 62.59)	0.013	27.48 ± 16.79	39.31 ± 43.97	11.83 (−6.63 to 6.25)	0.272	0.237

Results are expressed as ^+^ mean ± standard deviation (SD) (*n* = 38) and ^ǂ^ mean differences (95% confidence interval [CI]) between before and after each intervention. Before each intervention is the value of the previous intervention or the baseline in the first intervention. METS: metabolic equivalent. * *p*: Significant differences (*p* < 0.05) between before and after the intervention (intra-group changes). *p*-Value: Statistical differences between-group changes.

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
