# Peer review of "Consumption of Aged White Wine under a Veil of Flor Reduces Blood Pressure-Increasing Plasma Nitric Oxide in Men at High Cardiovascular Risk"

_nutrients, 2019, doi:10.3390/nu11061266_

Round 1
Reviewer 1 Report
The present study investigates the effects of Andalusian aged white wine (AWW) an alcoholic beverage with a moderate polyphenol content with those of gin on BP and plasma NO in men at high cardiovascular risk and demonstrates that in only 3 weeks only AWW decreased systolic and diastolic BP, as well as, increased the plasma NO concentrations suggesting that the hypotensive effects of AWW might be attributed to the non-alcohol compounds . However I suggest some major revisions:
· Among the clinical parameters considered, you also evaluated inflammatory markers (such as C-reactive protein) and serum creatinine as an index of renal function?
· At the end of the treatments, did you find any differences in the clinical blood profile of the subjects enrolled? If yes are there any differences between the group treated with AWW compared to those with gin?
· Have you evaluated or could you evaluate plasmatic markers of oxidative stress?
· To emphasize the importance of the redox balance as risk factor for cardiovascular diseases please integrate the discussion with these recent publications (PMID: 29379586; PMID 28495910; PMID 14744634; PMID 23905086)
Author Response
REVIEWER #1
The present study investigates the effects of Andalusian aged white wine (AWW), an alcoholic beverage with a moderate polyphenol content with those of gin on BP and plasma NO in men at high cardiovascular risk and demonstrates that in only 3 weeks only AWW decreased systolic and diastolic BP, as well as, increased the plasma NO concentrations suggesting that the hypotensive effects of AWW might be attributed to the non-alcohol compounds. However, I suggest some major revisions:
1. Among the clinical parameters considered, you also evaluated inflammatory markers (such as C-reactive protein) and serum creatinine as an index of renal function?
Comment1: Thank you for your comment. We have previously published the effects of AWW on these two parameters (and others) in “Roth I, et al. Clin Nutr. 2019 Jun;38(3):1036-1044”. In this previous study we measured changes in classical cardiovascular risk factors, cellular expression of circulating adhesion molecules, and endothelial progenitor cells (EPCs) and plasma biomarkers (CRP, IL-1β, IL-6, IL-8, IL-10, IL-12, IL-13, IFN-γ, MCP-1, TNF-α, IL-18, ICAM-1, VCAM-1) related to atherosclerosis after 3 weeks of intervention. A reference on the effects of AWW intervention on cellular and plasma inflammatory markers has been included in the Discussion (lines 248-263).
In addition, to determine the possible adverse effects of ethanol intake, plasma aspartate aminotransferases (AST), alanine transaminase (ALT), gamma glutamyl transpeptidase (GGT), albumin, serum creatinine as an index of renal function, cholesterol, prothrombin time, folic acid and vitamin B12 concentrations were also analyzed. No changes in these parameters were observed after AWW intake, suggesting that moderate consumption of this type of wine does not have relevant side effects. We have now included a reference on “creatinine” in the Results section (See line 173).
2. At the end of the treatments, did you find any differences in the clinical blood profile of the subjects enrolled? If yes are there any differences between the group treated with AWW compared to those with gin?
Comment 2: Again, thank you for your comment. In addition, to blood pressure, we also measured the effects of AWW and gin on other cardiovascular risk factors. Part of these results were published in a previous paper (Roth I, et al. Clin Nutr. 2019 Jun;38(3):1036-1044). In this analysis of the data, we observed no changes in body weight, body mass index or waist circumference throughout the trial. During this 3-week intervention, we also observed that the AWW intervention decreased systolic and diastolic BP by 3% and 4%, respectively (p = 0.033; both), and increased HDL-cholesterol concentrations by 9% (p = 0.001). There were no changes in these parameters after the gin intervention. On the other hand, gin intake was associated with increases in plasma glucose concentrations of 5% (p = 0.025), 0.7% (p = 0.001) in total cholesterol, 10% (p = 0.005) in LDL-cholesterol, and 9% (p = 0.001) in apolipoprotein B. Apolipoprotein AI concentrations and prothrombin time increased after both interventions (p ≤ 0.022; all). No significant changes were observed in CRP, creatinine, ASAT, ALAT, GGT, lipoprotein(a), thromboplastin time, fibrinogen, folic acid, vitamin B12, homocysteine, leukocytes, platelets, lymphocytes, monocytes, or hemoglobin A1c. Between group differences only revealed reductions in systolic and diastolic BP (p = 0.039; both) with AWW, while increases in GGT concentrations (p = 0.019) and prothrombin time (p = 0.038) were observed after gin consumption.
3. Have you evaluated, or could you evaluate plasmatic markers of oxidative stress?
Comment 3: Thank you for your suggestion. Unfortunately, because of a lack of funding, we did not evaluate oxidative stress markers in this study. As explained before, we only analyzed inflammatory biomarkers, since nowadays atherosclerosis is considered a low-grade inflammatory disease of the circulatory system. However, as oxidative stress is also key in the development of atherosclerosis and following the suggestion of the reviewer, we plan to evaluate oxidative stress markers in the near future. Taking this into account, we have included two paragraphs on this issue in the Discussion (lines 248-263) and a sentence considering this as a limitation of the study (lines 320-322).
4. To emphasize the importance of the redox balance as risk factor for cardiovascular diseases please integrate the discussion with these recent publications (PMID: 29379586; PMID 28495910; PMID 14744634; PMID 23905086)
Comment 4: Thank you for your suggestion. Accordingly, we have now included these publications in the Discussion (Lines 260-263). The changes have been highlighted in red.
Other studies have examined the relationship between hypertension and oxidative stress in experimental animals [26] and hypertensive subjects [27-28]. However, the results of these studies concluded that treatment with antioxidants is ineffective in hypertension, suggesting that oxidative stress is an effect (a consequence) rather than the cause of essential hypertension [26].

Reviewer 2 Report
This study investigates the effect of white wine consumption on BP and plasma NO in men. The idea behind the study is interesting and investigators have explained about the importance of this study; however, the study design needs a significant change and the paper needs to be corrected grammatically. Please see below my comments:
-Major grammar and punctuation issues throughout the entire paper.
-What is the justification for including only men in the study?
- Including smoking as one of the risk factors is correct; however, those who are smokers have different levels of inflammatory and oxidative stress compared to those who don't smoke. How would you justify that, considering that you are also measuring NO levels?
-What is your justification for choosing such a short period of time as the study duration?
-Total N of 38 for this study is very low and can cause issues with study power and the results.
-What is the rationale for choosing gin as the control arm? Gin is kind of on the heavier side as far as alcoholic beverages and its consumption may result in higher oxidative stress. Please explain how that can be a comparable control.
-Using paired t-test for a cross-over study design is not the best choice. Please consider re-analyzing your results using a different statistical method. Please add information on how the power calculation was done.
-Line 221-223: In order to draw such a conclusion, other measurements are required. For instance, measurement of isoforms important in NADPAH oxidase pathway, etc.
-Line 245: There is a clear gender effect based on that study, so please justify why you chose only men for your study.
-Overall, the study design needs to be modified. More measurements needs to be performed to draw conclusion about effect of AWW consumption on BP. For instance, there is no information on inflammatory markers, oxidative stress markers, etc. There should be justification on the inclusion criteria. There should be explanation on choice of control group. The result section needs to be more detailed.
Author Response
REVIEWER #2
This study investigates the effect of white wine consumption on BP and plasma NO in men. The idea behind the study is interesting and investigators have explained about the importance of this study; however, the study design needs a significant change and the paper needs to be corrected grammatically. Please see below my comments:
1. Major grammar and punctuation issues throughout the entire paper.
Comment 1: Thank you for this suggestion. Now, the paper has been revised by two native English speakers.
2. What is the justification for including only men in the study?
Comment 2: Metabolism of alcohol (ethanol) in the body differs between men and women. In fact, women show higher blood alcohol concentrations than men after the consumption of the same amount of alcohol. This has been attributed to: i. Different body size and composition (women have more fat than men –Bredella MA: AdvExp Med Biol 2017); ii: Different first-pass metabolism of ethanol due to its oxidation by gastric tissue (Frezza N, N Engl J Med 1990); iii. Different excretion rates by liver metabolism; and i.v. Different hormonal status (i.e. estrogen levels). Accordingly, recommendations on moderate alcohol consumption differ between men (up to 2-3 drinks a day) and women (up to 1-1½ drinks a day). Thus, we did not want to mix men and women in the same protocol. However, we agree with the reviewer that it would be very interesting to also test AWW in women.
3. Including smoking as one of the risk factors is correct; however, those who are smokers have different levels of inflammatory and oxidative stress compared to those who don't smoke. How would you justify that, considering that you are also measuring NO levels?
Comment 3: Thank you for your suggestion. The reviewer is right. Now we have included a subgroup analysis taking into account smoking and other cardiovascular risk factors. See lines 218 – 221 of the manuscript.
In the subgroup analyses, we did not find differences in the effects of AWW and GIN on BP and plasma NO concentrations in the different subgroups analyzed; that is, smoker vs non-smoker, diabetic vs. non-diabetic, hypertensive vs. non-hypertensive and dyslipidemic vs. non-dyslipidemic subjects.
4. What is your justification for choosing such a short period of time as the study duration?
Comment 4: Thank you very much for your comment. We have chosen a short period because of the crossover design, which also includes two washout periods. In a parallel-group design we agree that intervention periods should be longer (between 3 to 6 months). In our experience, 4 weeks of intervention is long enough to observe significant changes in almost all cardiovascular risk factors, except changes in body weight and waist circumference. On the other hand, 4 weeks also ensure high intervention compliance, as we have found in other trials performed by our group analyzing the effects of different food interventions on cardiovascular risk factor and inflammatory or oxidative stress biomarkers.
1. Vázquez-Agell M, Urpi-Sarda M, Sacanella E, Camino-López S, Chiva-Blanch G, Llorente-Cortés V, Tobias E, Roura E, Andres-Lacueva C, Lamuela-Raventós RM, Badimon L, Estruch R. Cocoa consumption reduces NF-κB activation in peripheral blood mononuclear cells in humans.NutrMetabCardiovasc Dis. 2013 Mar;23(3):257-63.
2. Chiva-Blanch G, Urpi-Sarda M, Ros E, Valderas-Martinez P, Casas R, Arranz S, Guillén M, Lamuela-Raventós RM, Llorach R, Andres-Lacueva C, Estruch R. Effects of red wine polyphenols and alcohol on glucose metabolism and the lipid profile: a randomized clinical trial.ClinNutr. 2013 Apr;32(2):200-6.
3. Chiva-Blanch G, Urpi-Sarda M, Ros E, Arranz S, Valderas-Martínez P, Casas R, Sacanella E, Llorach R, Lamuela-Raventos RM, Andres-Lacueva C, Estruch R. Dealcoholized red wine decreases systolic and diastolic blood pressure and increases plasmanitric oxide: short communication. Circ Res. 2012 Sep 28;111(8):1065-8.
4. Chiva-Blanch G, Condines X, Magraner E, Roth I, Valderas-Martínez P, Arranz S, Casas R, Martínez-Huélamo M, Vallverdú-Queralt A, Quifer-Rada P, Lamuela-Raventos RM, Estruch R. The nonalcoholic fraction of beer increases stromal cellderived factor 1 andthe number of circulating endothelial progenitor cells in high cardiovascular risk
subjects:a randomized clinicaltrial.Atherosclerosis. 2014; 233(2):518-24.
5. Chiva-Blanch G1, Urpi-Sarda M, Ros E, Arranz S, Valderas-Martínez P, Casas R, Sacanella E, Llorach R, Lamuela-Raventos RM, Andres-Lacueva C, Estruch R. Dealcoholized red wine decreases systolic and diastolic blood pressure and increases plasmanitric oxide: short communication. Circ Res. 2012 Sep 28;111(8):1065-8.
5. Total N of 38 for this study is very low and can cause issues with study power and the results.
Comment 5: We have assessed the sample size according to the SD of NO determination. Now we have included the following paragraph at the beginning of the statistical analysis in the Methods section (lines 148-151).
The ENE 3.0 statistical program (GlaxoSmithKline, Brentford, United Kingdom) was used to determine sample size. For a cross-over design and assuming a maximum loss of 10% of participants, to detect a mean difference of 4,1 µmol/L of NO with a conservative standard deviation (SD) of 6.0, at least 22 subjects are needed for the study (α risk = 0.05, power = 0.9).
6. What is the rationale for choosing gin as the control arm? Gin is kind of on the heavier side as far as alcoholic beverages and its consumption may result in higher oxidative stress. Please explain how that can be a comparable control.
Comment 6: Thank you very much for your comment. We also agree that gin is a heavy alcohol drink, but it helped us to test our hypothesis. We wanted to test whether the effects (either positive or negative) of AWW are due to its alcohol content or to the non-alcoholic compounds contained in this alcoholic beverage. Thus, we planned to compare the same amount of alcohol (30 g ethanol/day) using AWW - that contains alcohol and several other compounds, mainly polyphenols - and gin that only contains alcohol (the amount of polyphenols in gin is nearly zero). After analyzing the polyphenol content of several alcoholic beverages, the only two that did not have polyphenols were gin and vodka, and we decided to use gin since gin consumption is more common than vodka consumption in Spain.
7. Using paired t-test for a cross-over study design is not the best choice. Please consider re-analyzing your results using a different statistical method. Please add information on how the power calculation was done.
Comment 7: Thank you for your comment. In other trials performed by our group with a similar cross-over design, we analyzed different statistical strategies, mainly ANOVA analysis, and we decided (after consultation with different statisticians) on a two-tailed paired t-test, since no adjustments are needed. Each participant is their own control.
We have used this methodology in several manuscripts:
1. Vázquez-Agell M, Urpi-Sarda M, Sacanella E, Camino-López S, Chiva-Blanch G, Llorente-Cortés V, Tobias E, Roura E, Andres-Lacueva C, Lamuela-Raventós RM, Badimon L, Estruch R. Cocoa consumption reduces NF-κB activation in peripheral blood mononuclear cells in humans. NutrMetabCardiovasc Dis. 2013 Mar;23(3):257-63.
2. Chiva-Blanch G, Urpi-Sarda M, Ros E, Valderas-Martinez P, Casas R, Arranz S, Guillén M, Lamuela-Raventós RM, Llorach R, Andres-Lacueva C, Estruch R. Effects of red wine polyphenols and alcohol on glucose metabolism and the lipid profile: a randomized clinical trial. ClinNutr. 2013 Apr;32(2):200-6.
3. Chiva-Blanch G, Urpi-Sarda M, Ros E, Arranz S, Valderas-Martínez P, Casas R, Sacanella E, Llorach R, Lamuela-Raventos RM, Andres-Lacueva C, Estruch R. Dealcoholized red wine decreases systolic and diastolic blood pressure and increases plasmanitric oxide: short communication. Circ Res. 2012 Sep 28;111(8):1065-8.
4. Chiva-Blanch G, Condines X, Magraner E, Roth I, Valderas-Martínez P, Arranz S, Casas R, Martínez-Huélamo M, Vallverdú-Queralt A, Quifer-Rada P, Lamuela-Raventos RM, Estruch R. The non-alcoholic fraction of beer increases stromal cell derived factor 1 and the number of circulating endothelial progenitor cells in high cardiovascular risk subjects: a randomized clinicaltrial. Atherosclerosis. 2014 Apr;233(2):518-24.
1. Chiva-Blanch G1, Urpi-Sarda M, Ros E, Arranz S, Valderas-Martínez P, Casas R, Sacanella E, Llorach R, Lamuela-Raventos RM, Andres-Lacueva C, Estruch R. Dealcoholized red wine decreases systolic and diastolic blood pressure and increases plasma nitric oxide: short communication. Circ Res. 2012 Sep 28;111(8):1065-8.
8. Line 221-223: In order to draw such a conclusion, other measurements are required. For instance, measurement of isoforms important in NADPAH oxidase pathway, etc.
Comment 8: Thank you for your suggestion. Although it would have been very interesting to also evaluate oxidative stress markers in this trial, due to a lack of funding, we decided to focus on analyzing systemic inflammation rather than oxidative stress. This has now been stated as a limitation of the study (lines 317-320).
9. Line 245: There is a clear gender effect based on that study, so please justify why you chose only men for your study.
Comment 9: We agree with the reviewer that it would be very interesting to also study the effects of AWW in women. Please see above in the answer to question 2 the reasons why we only studied men.
10. Overall, the study design needs to be modified. More measurements need to be performed to draw conclusion about effect of AWW consumption on BP. For instance, there is no information on inflammatory markers, oxidative stress markers, etc. There should be justification on the inclusion criteria. There should be explanation on choice of control group. The result section needs to be more detailed.
Comment 10: Thank you for the suggestion. In our experience, the effects of food intervention (including wine) on cardiovascular risk factors are greater in subjects at high cardiovascular risk than in healthy volunteers (without cardiovascular risk factors). This is the main reason why we included subjects with moderate-high risk factors in the study. In fact, as stated in lines 169-171, most participants were overweight or obese (88%), more than a half had dyslipemia (54%), nearly three-quarters had hypertension (73%), had type-2 diabetes (21%), and 13% were smokers. Thus, the participants fulfilled the inclusion criteria described in the Method section (lines 73-80). As explained above, the effects of AWW on inflammatory biomarkers have been reported elsewhere (1,2 below) and the absence of analysis of oxidative stress biomarkers has been included as a limitation of the study. However, we believe that the statistical association between changes in BP and plasma NO concentrations are valid by itself, because of the plausibility of the results.
References:
1. Roth I, Casas R, Ribó-Coll M, Doménech M, Lamuela-Raventós RM, Estruch R. Acute consumption of Andalusian aged wine and gin decreases the expression of genes related to atherosclerosis in men with high cardiovascular risk: Randomized intervention trial. ClinNutr. 2018 Jul 20. pii: S0261-5614(18)31216-0. doi: 10.1016/j.clnu.2018.07.014.
2. Roth I, Casas R, Medina-Remón A, Lamuela-Raventós RM, Estruch R. Consumption of aged white wine modulates cardiovascular risk factors via circulating endothelial progenitor cells and inflammatory biomarkers. ClinNutr. 2019 Jun;38(3):1036-1044. doi: 10.1016/j.clnu.2018.06.001.

Round 2
Reviewer 1 Report
Good revisions!
I have no further request
Author Response
Thak you very much for you comments.
Reviewer 2 Report
Thank you for addressing all the comments and modifying the manuscript accordingly. The only comment that I have is to check for punctuations and few remaining grammar issues.
Author Response
Thank you for your comments. the manuscript has been reviewed by a professional English editing service again. We attach certificate to the editor.